# What Evidence Do We Have for Pharmaceutical Galactagogues in the Treatment of Lactation Insufficiency?—A Narrative Review

**DOI:** 10.3390/nu11050974

**Published:** 2019-04-28

**Authors:** Luke E. Grzeskowiak, Mary E. Wlodek, Donna T. Geddes

**Affiliations:** 1Adelaide Medical School, Robinson Research Institute, The University of Adelaide, Adelaide, SA 5005, Australia; 2SA Pharmacy, Flinders Medical Centre, SA Health, Bedford Park, Adelaide, SA 5042, Australia; 3Department of Physiology, The University of Melbourne, Melbourne, VIC 3010, Australia; m.wlodek@unimelb.edu.au; 4School of Molecular Sciences, The University of Western Australia, Crawley, Perth, WA 6009, Australia; donna.geddes@uwa.edu.au

**Keywords:** galactagogues, low milk supply, lactation, breast milk, breast feeding

## Abstract

Inadequate breast milk supply is a frequently reported reason for early discontinuation of breastfeeding and represents a critical opportunity for intervening to improve breastfeeding outcomes. For women who continue to experience insufficient milk supply despite the utilisation of non-pharmacological lactation support strategies, pharmacological intervention with medications used to augment lactation, commonly referred to as galactagogues, is common. Galactagogues exert their pharmacological effects through altering the complex hormonal milieu regulating lactation, particularly prolactin and oxytocin. This narrative review provides an appraisal of the existing evidence regarding the efficacy and safety of pharmaceutical treatments for lactation insufficiency to guide their use in clinical practice. The greatest body of evidence surrounds the use of domperidone, with studies demonstrating moderate short-term improvements in breast milk supply. Evidence regarding the efficacy and safety of metoclopramide is less robust, but given that it shares the same mechanism of action as domperidone it may represent a potential treatment alternative where domperidone is unsuitable. Data on remaining interventions such as oxytocin, prolactin and metformin is too limited to support their use in clinical practice. The review provides an overview of key evidence gaps and areas of future research, including the impacts of pharmaceutical galactagogues on breast milk composition and understanding factors contributing to individual treatment response to pharmaceutical galactagogues.

## 1. Mothers’ Own Breast Milk is Best

Breast milk is considered the optimal source of enteral nutrition to support the growth and development of all infants [1]. Exclusive breastfeeding is advocated for the first six months, with the continuation of breastfeeding supported until the age of two years or longer [2]. Infants who are not breastfed or are breastfed for shorter durations are at an increased risk of early and later-life morbidities and mortality [1,2,3]. Specifically, the reduced provision of breast milk is associated with increased risks of gastrointestinal, urinary, respiratory and middle-ear infections, together with a range of non-communicable diseases such as asthma, allergies, obesity and diabetes, and some childhood cancers [4]. Breastfeeding also confers a range of maternal benefits, which include reduced risks of breast, endometrial and ovarian cancer, together with faster return to pre-pregnancy weight and improvements in the psychological wellbeing [4,5]. 

The provision of mothers’ own breast milk is particularly important in the feeding of vulnerable preterm infants, where superiority of mothers’ own milk to donor human milk and infant formula with respect to the composition and bioactivity are critically important [3]. The use of mothers’ own breast milk during hospitalisation reduces the incidence and severity of preventable morbidities, including necrotizing enterocolitis (NEC), late onset sepsis, chronic lung disease, retinopathy of prematurity, rehospitalisation after discharge, and neurodevelopmental problems in infancy and childhood [3]. For example, a recent Cochrane review showed that preterm infants receiving infant formula are three times more likely to develop NEC, which has a mortality rate of 20–40% [6]. The benefits are further exemplified by evidence from a cohort of very low birth weight infants where for every 10 mL/kg/day increase in the ingestion of mothers’ own breast milk, the odds of rehospitalisation up to 30 months of age decreased by 5% [7]. These benefits extend long-term, with predominant breast milk feeding in the first 28-days of life is associated with better IQ, academic achievement, working memory, and motor function at seven years of age in very preterm infants [8]. In addition, the ability for a mother to provide her own breast milk provides psychological benefits including greater feelings of attachment, empowerment, and confidence [9]. 

The reality, however, is that many mothers face challenges related to initiating and sustaining breastfeeding. One of the key challenges related to breast milk production, with perceived inadequate breast milk supply is a frequently reported reason for early discontinuation of breastfeeding or decreased exclusivity in women who have initiated breastfeeding [10]. As an example of the potential magnitude of the problem, a large survey of 1323 mothers in the US highlighted that 50% of women stopped breastfeeding earlier than they had intended due to insufficient breast milk supply [11]. These surveys however, suffer a lack of objective evidence to confirm whether the low milk supply is real or perceived. Further, they often do not include information about early breastfeeding practices during the establishment of lactation such as the frequent emptying of the breast. Nevertheless, even if true lactation insufficiency is a fraction of previously reported figures, a significant opportunity exists for the development and implementation of strategies targeting improvements in breast milk supply to improve breastfeeding outcomes and subsequent maternal and infant health. In light of this, there continues to be widespread interest in and use of medications to augment lactation, commonly referred to as galactagogues. Prior to appraising the evidence regarding the role of pharmaceutical galactagogues in the treatment of lactation insufficiency, it is necessary to consider the physiology of lactation.

## 2. Physiology of Lactation

The development of functional lactation is a multi-stage event and is regulated by a complex hormonal milieu that includes numerous reproductive hormones (e.g., oestrogen, progesterone, prolactin, oxytocin) and metabolic hormones (e.g., glucocorticoids, insulin, insulin-like growth factor 1 (IGF-I), growth hormone, and thyroid hormone) [12]. Secretory differentiation occurs in mid- to late pregnancy when the differentiation of mammary epithelial cells into lactocytes occurs, conferring the ability to synthesize and secrete key components of human milk. This process is largely driven by circulating levels of oestrogen and progesterone that not only stimulate milk duct development, but also suppress prolactin’s action on milk production during pregnancy. Secretory activation is triggered by the sudden drop in progesterone following delivery of the placenta, accompanied by high levels of circulating prolactin. Secretory activation is confirmed by the onset of copious milk secretion and typically occurs between 24–102 h (average 60 h) after birth. This is often described as the milk ‘coming in’. Milk ejection is primarily regulated by oxytocin, which stimulates the contraction of myoepithelial cells around the alveoli and facilitates milk release [13]. Once lactation is established, an autocrine control (local feedback) mechanism regulates ongoing milk production, meaning supply is largely based on the effective removal of milk from the breast and is less sensitive to circulating levels of prolactin [13]. 

Given the important roles of prolactin and oxytocin in regulating milk secretion and milk ejection, these hormones have become common pharmacological targets for influencing breast milk supply. Figure 1 provides an overview of endogenous and exogenous factors influencing prolactin and oxytocin, while also highlighting mechanisms of action of common pharmaceutical galactagogues. A more detailed review of the neuroendocrine regulation of prolactin and oxytocin secretion and function can be found elsewhere and is beyond the scope of this review [14]. In brief, the production of prolactin in the anterior pituitary is regulated by the presence of prolactin inhibiting factors (PIF) and prolactin releasing factors (PRF), which are controlled by the hypothalamus. The concentrations of these inhibitory and stimulatory factors is influenced by a variety of external stimuli such as infant suckling, sounds of the infant crying, and stress. The key PIF is dopamine, which exerts an inhibitory effect on prolactin production. Positive external stimuli lead to an inhibitory effect on dopamine release and a resultant increase in prolactin levels. Similarly, medications that block dopamine, such as the dopamine antagonists domperidone and metoclopramide, block its inhibitory effects and result in an increase in prolactin secretion. In contrast, a range of hormones act to stimulate prolactin production, including thyrotropin releasing hormone, cortisol, and oxytocin. Other hormones that exert direct or indirect effects on the mammary gland to influence milk synthesis include the growth hormone, prolactin and insulin. Milk contains a small whey protein called the feedback inhibitor of lactation (FIL). The accumulation of milk in the breast leads to milk stasis and an increased concentration of FIL, which provides a negative feedback to slow milk production by inhibiting the action of prolactin on lactocytes. 

## 3. Risk Factors for Lactation Insufficiency

Three key aspects are required for establishing an adequate breast milk supply including sufficient mammary tissue, normal hormone levels, and regular effective removal of milk [17]. Insufficient mammary tissue can be the result of hypoplastic breasts, breast surgery such as mastectomy or breast reduction, or cyst removal [18]. Numerous maternal factors can influence hormone levels including retained placenta, severe postpartum haemorrhage, hypothyroidism, high levels of stress or anxiety, certain medications, anaemia, diabetes, obesity, polycystic ovary syndrome, cigarette smoking, or alcohol consumption [17,18]. Lastly, the effective and regular removal of milk can be influenced by the method and frequency of expressing if indirectly feeding the infant, or the inability for infants to effectively remove milk from the breast if feeding directly as a result of issues such as poor attachment, low infant intraoral vacuums and delivery complications. Factors such as the use of infant formula, early introduction of solids, infant sedation, or infant medical problems (e.g., low birth weight, congenital abnormalities, illness) may also exacerbate poor feeding and ineffective removal of breast milk [18]. 

In particular, mothers of preterm infants face many challenges in establishing and maintaining an adequate supply of breast milk during their infant’s prolonged hospitalisation. This is driven by multiple factors including; physiological immaturity of the breast associated with preterm birth, maternal morbidities, inability for the preterm infant to breastfeed directly from the breast, and the stress of having an infant admitted to the Neonatal Intensive Care Unit (NICU). Each of these factors has the ability to interfere with the establishment of normal milk supply, with one previous study demonstrating that 82% of women birthing preterm experienced delayed secretory activation [19]. While longer-term breastfeeding outcomes were not collected for this cohort, other studies in mothers of term infants have demonstrated that delayed secretory activation is associated with an increased risk of early cessation of breastfeeding [20].

## 4. Use of Galactagogues in Clinical Practice

Relatively little evidence exists regarding the prevalence of pharmaceutical galactagogue use. A recent clinical practice survey of Australian Neonatal Units (NNUs) identified that 100% of NNUs reported domperidone as their first line pharmacological agent of choice in the management of lactation insufficiency [21]. While 75% of NNUs reported having a clinical guideline on domperidone, significant variability was evident across these guidelines with respect to the dose and duration of treatment. 

The preference towards the use of domperidone for lactation insufficiency is supported by a recent audit of phone calls to an Australian pregnancy and lactation counselling service regarding the use of galactagogues during lactation highlighted that domperidone accounted for >90% of phone calls [22]. In contrast, the discussion of herbal galactagogues increased from 0% in 2001 to 23% in 2014. Overall, these findings highlight the popularity of domperidone use in the Australian setting, together with the increasing popularity of herbal galactagogues. 

Three studies have examined patterns of domperidone use over the past 10–15 years. An Australian study undertaken at a single tertiary level maternity hospital identified the increasing prevalence of use from 0.1% of all deliveries in 2000 to 5% in 2010 [23]. A recognised limitation of this study lies in the fact that it likely underestimates total domperidone use, as it could not take into account domperidone dispensed from community pharmacies. In comparison, a Canadian study investigated the number of women that dispensed domperidone in the first six months postpartum among 320,351 live births occurring between 2002 and 2011 [24]. Over this time period the prevalence of domperidone use increased from 7% to 18% in women delivering at term and 17% to 32% in women delivering preterm. The most recent study used data from the Clinical Practice Research Datalink in England and demonstrated a 3.8-fold increase in postpartum domperidone prescribed from 0.56 per 100 person-years in 2002–2004 to 2.1 per 100 person-years in 2011–2013, with overall use much lower than that reported in previous studies [25]. Risk factors commonly associated with domperidone use in these studies included preterm birth, delivery by caesarean section, maternal obesity, diabetes or gestational diabetes, and primiparity [24,25,26]. 

## 5. Efficacy and Safety of Galactagogues

In order to evaluate the efficacy and safety of galactagogues we searched four electronic databases from inception to January 2019: Ovid Medline, EMBASE, Web of Science, and SCOPUS. Medical subject headings (e.g., MeSH headings) and free word combinations using Boolean logic of the following search items were used: ‘galactagogues’ OR individual medication names (e.g., ‘domperidone’) AND ‘lactation’ OR ‘low milk supply’. Previous reviews, bibliographies of published trials and cross references were also searched. No language restrictions were applied. For the evaluation of galactagogue efficacy, studies were restricted to those investigating the use of galactagogues in the treatment of established lactation insufficiency and using controlled study designs. Studies investigating herbal galactagogues were excluded from this review.

An overview of controlled intervention trials evaluating the efficacy of various galactagogues is provided in Table 1. Domperidone [27,28,29,30,31,32,33,34,35,36,37] and metoclopramide [28,29,38,39,40] appear to be the most commonly studied galactagogues in the setting of established lactation insufficiency, with limited studies evaluating the effects of other galactagogues including sulpiride [41,42], growth hormone [43,44], human recombinant prolactin [45], thyrotropin releasing hormone [46,47], and metformin [48]. No studies have evaluated the use of oxytocin in the treatment of lactation insufficiency. The focus of this review is on the use of galactagogues in the treatment of lactation insufficiency, studies that included women without an established diagnosis of lactation insufficiency were not considered [49,50,51,52,53,54,55,56,57,58,59]. 

### 5.1. Domperidone

Domperidone is a dopamine antagonist and is thought to act as a galactagogue by increasing serum prolactin (Figure 1). The majority of studies evaluating the effects of domperidone have been undertaken among mothers of preterm infants. Treatment durations (five to 28 days) and treatment initiation (>1 to >3 weeks post-partum) varied widely between studies. The most commonly used interventional dose consisted of 10 mg three times daily, with two studies evaluating a higher dose of 20 mg three times daily [27,34].

All placebo-controlled studies demonstrate treatment effects in favor of domperidone. Five studies involving mothers of preterm infants have recently been combined in a meta-analysis, demonstrating that the administration of 30 mg/day led to a modest increase in maternal breast milk volume of 88 mL daily [60]. Missing from this meta-analysis was the recent study by Fazilla et al., which demonstrated a similar improvement to the previous pooled estimate of 109 mL/day (95%CI 69 to 149 mL/day) [32]. Effects of domperidone on mothers of term infants is limited to two studies [33,35], with both demonstrating treatment effects in favor of domperidone compared with placebo. Inam et al. utilised a treatment target of 50 mL per single expression episode (both breasts) with 36 of 50 (72%) women in the domperidone arm achieving this target compared with 11 of 40 (22%) women in the placebo arm (*p* = 0.002) [33]. Petraglia et al. evaluated changes in breast milk supply by weighing infants before and after breast feeding, demonstrating greater improvement in daily breast milk supply among women receiving domperidone (347 ± 36 to 673 ± 44 mL/day), compared with placebo (335 ± 30 to 398 ± 45 mL/day; *p* < 0.01) [35].

Two underpowered studies have compared the effects of different doses of domperidone [27,34]. One of these was a small case-crossover study where women were randomised to receive an initial dose of 30 mg/day or 60 mg/day, then crossover to the other arm of the study after 7–14 days of treatment. Daily milk volume production was only reported for four of six women who were considered treatment responders, with daily breast milk volume increasing from 208.8 ± 182.3 mL/day at baseline to 566.4 ± 229.3 mL/day at a dose of 30 mg/day and 705.6 ± 388.1 mL/day at a dose of 60 mg/day [27]. The reported increases were statistically significantly higher compared with baseline values, but the small sample size of four women prohibited the identification of statistically significant differences between the two different doses. In contrast, Knoppert et al. randomised 15 women to either receive 30 mg/day or 60 mg/day for four weeks [34]. Daily breast milk volume increased from a median of 150 to 420 mL/day among those receiving the 30 mg/day dose, and 300 to 720 mL/day among those receiving the 60 mg/day dose, but the differences between the groups were not statistically significant [34]. Collectively, while these two studies provide promising evidence of potential further improvements in daily breast milk volume with the use of a higher dose of domperidone (i.e., 60 mg/day compared to 30 mg/day), further research is required to determine if a higher dose can be considered superior. 

Maternal adverse events most commonly associated with the use of domperidone in clinical trials include headache, dry mouth, and gastrointestinal disturbances. That said, compared to placebo, no increased risk of maternal adverse events (RR 1.05; 95%CI 0.65–1.71) has been identified in clinical trials. Similarly, no trials have reported an increased risk of adverse neonatal events [58,59]. Indeed the concentrations of domperidone in breast milk are low with the absolute infant dose only 0.04 (95%CI 0.03–0.07) micrograms/kg/day for a maternal dose of 30 mg/day, and 0.07 (95%CI 0.05–0.11) micrograms/kg/day for a maternal dose of 60 mg/day [27]. One small case-crossover study demonstrated a higher prevalence of adverse events with a higher dose of domperidone [27]. Among the seven study participants, there were a total of five adverse events reported at 30 mg/day (abdominal cramping [*n* = 1], dry mouth [*n* = 3], headache [*n* = 1]), and 12 adverse events reported at 60 mg/day (abdominal cramping [*n* = 2], constipation [*n* = 1], dry mouth [*n* = 5], depressed mood [*n* = 1], headache [*n* = 3]) [27]. Also, one woman withdrew from the study as a result of experiencing severe abdominal cramps while receiving 60 mg/day [27]. In contrast, a more recent double blind randomised controlled trial comparing 30 mg/day to 60 mg/day identified no adverse events in either group [34]. Therefore, while it currently remains unclear, it is possible that a higher dose of domperidone may be associated with a greater risk of maternal adverse events.

The use of domperidone in lactation has been the subject of controversy due to an increased risk of ventricular arrhythmia (VA) and sudden cardiac death of approximately four per 1000 person-years observed among non-lactating adults, including males and females [61,62]. This increased risk relates to the potential of domperidone to prolong the QT interval [63], but the relevance to lactating women has been questioned [64,65,66,67,68]. Domperidone was initially developed as an antiemetic and prokinetic, with previous recommended doses ranging from 30 mg/day to 60 mg/day or greater [66]. Some observational studies suggested that the risk of VA or sudden cardiac death may be increased if daily doses were greater than 30 mg or if patients were male or greater than 60 years old [61,62]. More detail regarding domperidone and QT interval effects among non-lactating adults can be found elsewhere [61,62]. 

Numerous studies provide reassuring evidence as to the safety of domperidone use in lactating women. A recent randomised double-blind, placebo- and positive-controlled safety study identified that among healthy female volunteers, domperidone at doses up to 80 mg/day did not cause clinically relevant QTc-interval prolongation [69]. Further, all women participating in the trial by Asztalos et al. had an ECG at baseline and at the end of the study. They observed no evidence of prolongation of the QTc-interval with any mother participating in the trial [62]. Perhaps the strongest evidence to guide the clinical practice comes from a recent Canadian population-based cohort study of 320,351 women, of which 45,163 were prescribed domperidone within six months postpartum [70]. The primary outcome consisted of ventricular arrhythmia and data were obtained on whether or not women had a previous history of ventricular arrhythmia. No cases of ventricular arrhythmia were identified among those women with no prior history [71]. All cases of ventricular arrhythmia were observed among women with a previous history. Taken together, these results provide supporting evidence for the limited increased risk of ventricular arrhythmia among women with no previous medical history, and strengthens the support that domperidone should be avoided in those women with a previous history. Notably, among this cohort 90% of women received a dose of domperidone greater than 30 mg/day, suggesting limited impact of dose on ventricular arrhythmia risk. In light of such evidence it appears that domperidone can be used safely, but should be avoided or used with extreme caution in women with established risk factors for QTc-prolongation. This includes women taking medications that inhibit the metabolism of domperidone and/or also prolong the QTc-interval (e.g., fluconazole, erythromycin), and women with a personal or family history of cardiac arrhythmia or family history of unexplained sudden death. Clinical protocols regarding the use of domperidone in clinical practice have been developed and serve to facilitate safer prescribing practices and minimize potential adverse reactions in mothers and their hospitalized premature infants [72].

### 5.2. Metoclopramide

Similar to domperidone, metoclopramide is a dopamine antagonist and is thought to act as a galactagogue by increasing serum prolactin (Figure 1). In contrast to domperidone, the majority of clinical studies have evaluated the use of metoclopramide among mothers of term infants. Four studies have evaluated a standard dose of 10 mg three times daily, with the treatment duration varying from five to 21 days. One study evaluated three difference metoclopramide doses [39]. Three studies directly measured changes in breast milk volume, one study indirectly evaluated breast milk volume based on infant weight measures before and after a single feed, while the last study evaluated infant weight gain. A dose-response case-cross-over study identified a significant increase from baseline in breast milk yield during a single feed following use of 30 mg/day (42.5 ± 34.7 mL) and 45 mg/day (50.0 ± 35.9 mL) doses, but not 15 mg/day (11.2 ± 28.1 mL) or placebo (4.0 ± 27.5 mL) [39]. Blank et al. identified a greater increase in breast milk volume in those receiving metoclopramide compared with placebo (84 mL/day compared to 29 mL/day, but the difference between the groups was not statistically significantly different [28]. Kauppila et al. identified a significant increase in breast milk supply during the study among those receiving metoclopramide (285 ± 75 mL /day to 530 ± 162 mL /day (*p* < 0.01)), but change in breast milk supply was not reported for those in the placebo arm [38]. 

Adverse events associated with metoclopramide have been incompletely described. Kauppila et al. reported adverse events occurring among six of eight women, which included tiredness (*n* = 6), headache (*n* = 1), and nausea (*n* = 1) [38]. In the dose-response study by Kauppila et al. seven women complained of side-effects including tiredness (*n* = 1), headache (*n* = 1), anxiety (*n* = 1), hair loss (*n* = 1) and intestinal disorders (*n* = 1), but the relationship between the adverse events and dose were not described [39]. While greater amounts of metoclopramide are secreted into the breast milk compared with domperidone, the reported absolute infant dose ranging from six to 24 micrograms/kg/day is still well below the recommended pediatric dose of 500 micrograms/kg/day [73]. Notably, metoclopramide was detected in the plasma of one infant and while no short-term neonatal adverse events have been identified, whether infants are more susceptible to potential longer-term adverse events of metoclopramide is unknown. This is of concern given the ease with which metoclopramide crosses the blood-brain barrier and therefore the potential to interrupt dopamine signaling in the newborn.

Similar to domperidone, metoclopramide also has the potential to cause serious maternal adverse events. Given that metoclopramide has the ability to cross the blood-brain barrier, it is more likely to cause central nervous system side effects than domperidone. Of particular concern is the increased risk of depression, which is already increased in the postpartum period. In the study by Ingram et al., two of 29 (7%) women receiving open-label metoclopramide after the initial 10-day trial reported experiencing depression [29]. Metoclopramide also has the ability to cause serious and potentially permanent extrapyramidal side effects such as tardive dyskinesia, for which it carries a black box warning in the product information. As such, it is recommended that metoclopramide not be used for longer than five days and that the maximum dose be limited to 10 mg three times daily. A large international internet survey of women who took metoclopramide to increase breast milk supply found that 4.8% of women had either palpitations or racing heart rate, 12% reported depression, and 1 to 7% reported other central nervous system side effects ranging from dizziness and headache to involuntary grimacing and tremors [74]. The overall adverse event profile of metoclopramide makes it less desirable than domperidone to use as a galactagogue, with evidence directly comparing the two outlined below. 

### 5.3. Domperidone Compared To Metoclopramide

While domperidone and metoclopramide are both dopamine antagonists and increase serum prolactin concentrations, they do differ with respect to some pharmacodynamic properties. One of these key properties relates to the ability to cross the blood-brain barrier, with only metoclopramide able to do so in appreciable quantities. This has led to questions regarding potential differences in the efficacy and safety of these two medications. Two studies have directly compared outcomes among women taking domperidone and metoclopramide. Blank et al. identified a slightly greater increase in the daily breast milk volume among nine women receiving domperidone (120 ± 81 mL/day to 239 ± 105 mL/day) than among eleven women receiving metoclopramide (100 ± 53 mL/day to 184 ± 100 mL /day) [28]. The 35 mL/day difference between those receiving domperidone and metoclopramide, however, was not statistically significant. Ingram et al. demonstrated a similar difference in breast milk volume for those receiving domperidone compared with metoclopramide (31.0 mL/day; 95%CI −5.67 to 67.6 mL/day), among a total sample of 65 women [29]. 

When it comes to maternal adverse events, Blank et al. reported that minor neurobehavioral symptoms of either drowsiness, sleep disturbance, restlessness or dizziness were experienced by two of nine women taking domperidone and two of 11 women taking metoclopramide [28]. In contrast, Ingram et al. reported that seven (20%) of women taking metoclopramide reported experiencing adverse events compared with just three (10%) of women taking domperidone. Notably, one woman taking metoclopramide withdrew from the study due to experiencing bad headaches and dry mouth [29]. Greater incidence of adverse events with metoclopramide is also evident from observational studies, with a 2010 Internet survey of 1990 mothers identifying a 7-fold increased risk of depression, and 4 to 19-fold increased risk of symptoms commonly associated with tardive dyskinesia (e.g., tremors, involuntary grimaces, and jerking) among women taking metoclopramide compared with domperidone [75].

### 5.4. Sulpiride

Sulpiride is a dopamine antagonist, traditionally used as an antipsychotic, and is thought to act as a galactagogue by increasing serum prolactin (Figure 1). The effects of sulpiride in increasing breast milk supply have been evaluated in two placebo-controlled studies [41,42]. Both utilised a dose of 50 mg three times daily compared with placebo, with the treatment duration ranging from 14 to 28 days. It is unclear from both studies as to the infant gestational age. Both studies demonstrated a greater increase in breast milk volume among those receiving sulpiride, with improvements of 218 mL/day and 315 mL/day and associated reductions in the requirement for supplemental feeds. However, both studies experienced greater than 20% loss to follow-up in the placebo group which was reported to be due to insufficient treatment response, indicating significant potential for bias. While previous clinical trials in lactation have reported few adverse events associated with the use of sulpiride and the doses used in these trials are lower than those thought to produce significant neuroleptic effects in adults, data are still limited and there remains concerns regarding potential maternal or infant effects. Common adverse events include sedation and weight gain, while sulpiride may also cause extrapyramidal side effects similar to metoclopramide. Of greater concern is that the relative infant exposure to sulpiride through breast milk is up to 20% of the maternal weight-adjusted dose [49], which is much greater than that for other galactagogues. 

### 5.5. Growth Hormone

The exact mechanisms of action of growth hormone in lactation remain unclear. The growth hormone is released by the anterior pituitary gland and acts synergistically to prolactin [12]. Administration of the growth hormone has been demonstrated to increase secretion of the insulin-like growth factor, providing evidence that the galactopoietic response to the growth hormone occurs as a result of indirect effects on mammary glands [44]. Two studies have evaluated the administration of the human growth hormone [43,44]. The first placebo-controlled study enrolled mothers of preterm infants (<34 week’s gestation) and involved a subcutaneous administration of 0.2 international units/kg/day for seven days. Those receiving growth hormone experienced an increase in breast milk volume (139 ± 49 to 175 ± 46 mL/day; *p* < 0.01), but no difference was evident among those receiving placebo (93 ± 50 to 102 ± 69 mL/day; *p* = not stated). The second study, by the same authors, enrolled mothers of term infants and investigated three doses ranging from 0.05 to 0.2 international units/kg/day for seven days [44]. Women in the high dose arm (0.2 IU) experienced a greater percentage increase in breast milk volume than those in the low- (0.05 IU) and mid- (0.1 IU) dose arms (36 ± 12.6% compared to. 4.7 ± 9.7%; *p* < 0.04) [44]. Absolute differences in milk volume identified in the two previously reported studies were 25 and 86 mL/day, respectively. No maternal or infant adverse events were reported within either study. Overall, the relatively modest improvements in breast milk supply observed with growth hormone do not seem to justify the high costs and administration challenges associated with the treatment, particularly in comparison to alternative galactagogues such as domperidone or metoclopramide.

### 5.6. Recombinant Human Prolactin

Given that prolactin is a critical component of establishing lactation, the direct administration of prolactin is viewed as an approach for treating lactation insufficiency that is the result of prolactin deficiency (Figure 1). One study that involved the administration of recombinant human prolactin (r-hPRL) was identified [45]. The placebo-controlled study enrolled mothers of preterm infants (<32 weeks gestation) with prolactin deficiency, and involved subcutaneous administration of either 60 micrograms/kg twice daily or 60 micrograms/kg once daily for seven days. The subgroup analysis revealed that milk volumes were higher in the group treated with twice daily r-hPRL (429 ± 338%) than in the group receiving only placebo (−12 ± 27%; *p* < 0.05). No significant differences were observed with respect to maternal or infant adverse events. While data are limited, this study demonstrates that r-hPRL may be a viable option for the treatment of lactation insufficiency among women with established prolactin deficiency. Given the high costs and administration challenges associated with the treatment, this remains a treatment for rare situations where alternative prolactin stimulating medications such as domperidone are unsuitable, as is the case with the Sheehan syndrome or congenital prolactin deficiency where lactotrophs are absent or reduced in number. 

### 5.7. Thyrotrophin Releasing Hormone

The thyrotrophin-releasing hormone (TRH) acts on the anterior pituitary to increase serum prolactin concentrations (Figure 1). Two studies were identified that involved the administration of TRH, producing mixed results [46,47]. One study randomised women to receive either TRH (*n* = 5) or placebo (*n* = 4) orally [47]. Among women treated with TRH, only one woman experienced an increase in serum prolactin. Actual breast milk volumes were not described, with the effect on milk production reported as ‘no change’ for all women among the treatment and placebo arms of the study. The second study enrolled mothers of term infants and involved the intranasal administration of 1 mg four times daily for 10 days. Those receiving TRH experienced an increase in breast milk volume (142.0 ± 33.9 g/day to 253.0 ± 105.3 g/day; *p* = 0.014), but no difference was evident among those receiving placebo (150.0 ± 46.2 g/day to 140.6 ± 57.7 g/day; *p* = 0.87). The increase in breast milk volume was accompanied by an observed increase in basal prolactin levels from a mean of 117 (± 45) micrograms/L to 173 (± 56) micrograms/L among women receiving TRH, whereas those receiving placebo experienced a decrease in basal prolactin levels (137 ± 70 micrograms/L to 82 ± 38 micrograms/L). No maternal or infant adverse events were reported in these two studies. However, reported effects on thyroid function have been mixed based on open-label non-controlled studies, with one demonstrating no changes in maternal or infant thyroid function [47], while one study identified two cases of maternal hyperthyroidism among 13 women treated with TRH which required treatment discontinuation [59]. Given the potential severity of adverse events and mixed findings of previous studies, more research on the use of TRH is necessary before its consideration in clinical practice.

### 5.8. Oxytocin

Despite the important role of oxytocin in regulating milk ejection, no studies have evaluated the use of oxytocin in the setting of established lactation insufficiency. Existing literature is restricted to studies enrolling women soon after delivery, with treatment undertaken in an effort to augment early milk supply and prevent the development of lactation insufficiency [53,54,56,57,58]. These studies have produced conflicting findings and their relevance to the treatment of women with established lactation insufficiency is uncertain. As it currently stands, there is insufficient evidence whether oxytocin may be an effective treatment for lactation insufficiency. However, given evidence that acute or chronic stress can interfere with oxytocin secretion, inhibiting both milk transfer and mother-infant bonding [20], further research on the potential usefulness of oxytocin in the setting of established lactation insufficiency is warranted, particularly among mothers of preterm infants where high levels of anxiety and depressive symptoms are common. 

### 5.9. Metformin

Metformin is commonly used in the management of type 2 diabetes and is thought to act as a galactagogue through its role as an insulin-sensitising medication (Figure 1). The effects of metformin in increasing breast milk supply have been evaluated in one placebo controlled study [48]. The pilot study enrolled mothers of term infants who were experiencing low breast milk production and who had at least one sign of insulin resistance (e.g., elevated fasting plasma glucose). Women were randomised to either metformin (*n* = 10) or placebo (*n* = 5) with metformin dose increasing from 750 mg/day to 2000 mg/day over a 28-day treatment period. Peak median change in the milk output did not differ between those receiving metformin (+8; IQR –23 to 33 mL/day) or placebo (–58; IQR –62 to –1 mL/day) (*p* = 0.31). Notably, a greater increase in milk output was evident among those who managed to continue taking metformin for the entire treatment period, but only 20% of women assigned to the metformin arm experienced any actual improvement in milk output to day 28. Gastrointestinal adverse events were commonly reported, including diarrhoea (56%), nausea/vomiting (44%), and abdominal pain or cramping (56%), but were mostly of mild or moderate severity. One woman discontinued metformin due to the severity of abdominal cramping. No infant adverse events were reported. Given evidence on metformin is restricted to a small pilot study, its potential role as a galactagogue remains uncertain. 

### 5.10. Summary of Galactagogue Efficacy and Safety

The largest and strongest body of evidence supports the use of domperidone as the first-line medication for the pharmacological management of lactation insufficiency. Domperidone represents a low cost treatment that is well tolerated, with moderate quality evidence that it produces a modest increase in breast milk supply in mothers of preterm infants. Whether results are translatable to mothers of term infants remain unclear as there are a limited number of high quality studies that include this patient population. Domperidone has been associated with serious adverse events and therefore is not suitable for all women, however, following appropriate screening prior to prescribing the risk of serious adverse events appear negligible. There is weak evidence that domperidone may be slightly more effective than metoclopramide and that adverse events are more likely to occur with metoclopramide than domperidone. Metoclopramide may be an appropriate alternative for use in women where domperidone is unsuitable. Antipsychotics do not reflect a viable therapeutic option for the management of lactation insufficiency, largely owing to their side effects such as extrapyramidal reactions and weight gain. Limited evidence and difficulties with respect to access to and administration of medications such as growth hormone, recombinant human prolactin and thyrotrophin-releasing hormone make these less desirable treatment options. Lastly, while an initially promising therapy for improving breast milk supply among women with signs of insulin resistance, initial findings from a pilot study evaluating metformin do not provide compelling evidence to support its use as a therapeutic treatment of lactation insufficiency.

## 6. Methodological Challenges and Evidence Gaps Regarding Galactagogues

### 6.1. Previous Study Limitations

Clinical studies evaluating the use of pharmaceutical galactagogues suffer from a number of important methodological limitations. Highlighting these limitations is not only important in considering the quality of existing evidence, but also in informing future research endeavors. The major limitation of the existing evidence lies in the small number of women included in clinical trials, limiting the potential to comprehensively evaluate both the efficacy and safety of these medications. Studies ranged from 7 to 100 participants, with a median of only 24. Many studies were therefore underpowered to detect statistically or clinically significant differences in breast milk supply or adverse events. Further, smaller studies face greater challenges in achieving balance in the distribution of baseline covariates across study groups. For example, in the study by Nommsen-Rivers et al., six out of 10 (60%) women receiving metformin were also taking the herbal galactagogue fenugreek, compared with one out of five (20%) women in the placebo arm [48]. In the study by Knoppert et al., five out of seven (71%) women in the high dose domperidone arm were primiparous, compared with three out of eight (38%) in the low dose domperidone arm [34], whereas in the study by da Silva et al. six out of nine (67%) women in the placebo arm were primiparous, compared with six out of seven (86%) in the domperidone arm [31]. Such differences in the baseline distribution of factors potentially related to breast milk volume or treatment response to galactagogue can significantly bias study findings and highlights the importance of large, appropriately controlled trials. 

Many studies suffer from potential selection bias due to the high loss to follow-up in one or both treatment arms. This is particularly important where the loss to follow-up is related to the treatment response and only including data on those successfully completing the study potentially biases the results towards those with more favorable treatment response. Greater than 10% loss to follow-up in one or both intervention arms was evident among nine studies [28,29,31,34,36,37,39,41,42].

Non-pharmacological breastfeeding strategies are the mainstay of treatment for lactation insufficiency, but their implementation and utilisation across clinical trials of galactagogues is often incompletely described. This makes it unclear what strategies have been utilised and whether these have been applied equally across treatment groups. 

Two studies reported the encouragement of skin-to-skin contact between the mother and infant [29,37], while some provided advice regarding the minimum number of times women expressed each day, ranging from four to eight however the frequency of pumping was not accounted for despite strong relationships with the milk volume [28,29,34,37,43,48]. All studies where mothers were expressing breast milk reported the use of electric pumps, which have been shown to be more effective in the establishment of milk production than hand expression [75]. 

Definitions for lactation insufficiency varied widely across studies and involved the use of an absolute volume measure (e.g., <500 mL/day) [27,28,33,34,38,44,45], relative volume measure (e.g., < 250 mL/kg/day) [29,35,37,39,41,42,46], requirement for supplemental feeds [30,31,32,43], insufficient infant weight gain [39], or certain reduction in supply based on previous milk production [14,23,37]. Given that there is no universally accepted definition of lactation insufficiency in the literature, it is unsurprising that definitions used across studies vary greatly. It is uncertain, however, whether the baseline milk volume has any impact on the treatment response to pharmaceutical galactagogues and this could possibly explain some heterogeneity across clinical studies. Linked with this concept is the complete absence of any noted targets regarding the optimal breast milk volume in previously published clinical trials. This is particularly important for mothers of preterm infants where the inclusion of an optimal treatment target could have an impact on the treatment response. For example, if the target is simply defined as meeting the infants’ daily feed requirements and the infant is only 1 kg, the target is approximately 150 mL. As the infant grows, however, that target will shift and women whose daily breast milk production remains at 150 mL/day will find it difficult to keep up with their infants’ demands and this may lead to eventual cessation of breastfeeding. As an example, Meier and Engstrom have suggested that for mothers of very low birth weight infants a desirable milk volume target is at least 350 mL/day by the end of the second postnatal week [76]. Higher volumes (e.g., 600 to 800 mL/day) are considered even more desirable and are more likely to sustain long-term breastfeeding outcomes [76]. Among the studies included in this review, the median breast milk volume increased from 146 mL/day to 276 mL/day, with only five studies reporting a final mean volume above the proposed target of 350 mL/day [27,30,34,35,38], with all involving the use of either domperidone or metoclopramide. This indicates that the hormonal manipulation utilising existing therapeutic approaches may not be adequate to reverse the poor development of the mammary gland during pregnancy and milk synthesis after it is has been established.

While galactagogues may provide short-term improvements in breast milk supply, their relationship to longer-term breastfeeding outcomes is unknown. Long-term outcomes are particularly challenging to assess among placebo controlled trials as once the trial has completed many women are often able to take the intervention medication off-label. This means long-term outcomes become confounded by maternal behaviors following completion of the trial. With that in mind, few studies have reported long-term breastfeeding outcomes. Asztalos et al. provided the most comprehensive evaluation, with 60% of women still breastfeeding once their infants reached the term gestation corrected age, with 70% requiring additional supplementation [37]. Notably, 67% of women were still using lactation-inducting compounds following completion of the study. A major criticism of previous studies is that the majority only compare differences in supply between groups, not adjusting for baseline milk production. Adjusting for baseline values has been demonstrated to significantly improve statistical power, particularly where there is a strong correlation between baseline and final value [77].

Lastly, while no specific methodological limitation of the studies were included in this review, it is worthwhile to comment on the large number of studies that have investigated the use of pharmaceutical galactagogues in the setting where lactation insufficiency has not been diagnosed [49,50,51,52,53,54,55,56,57,58]. We do not feel that one can accurately draw evidence on the efficacy or safety of galactagogues where they are utilised in the absence of a clear established clinical indication. There is currently no evidence to support the prophylactic administration of galactagogues in the absence of the diagnosis of lactation insufficiency, while an associated concern with the use of galactagogues is that they may come at the cost of appropriate utilisation of non-pharmacological treatment strategies. 

### 6.2. Impact of Galactagogues on Breast Milk Composition

Prolactin is thought to mediate changes in breast milk composition occurring during normal lactogenesis [78]. In vitro and animal studies provide evidence that in early lactation prolactin promotes the closure of epithelial tight junctions between alveolar cells and increases the synthesis of α-lactalbumin, which in turn increases breast milk volume [78]. Prolactin has also been demonstrated to influence secretion of immunomodulatory factors in breast milk, with the decreasing concentrations of these factors occurring over the course of lactation mirroring reductions in serum prolactin concentrations also occurring over time [78]. Whether galactagogues, particularly those altering prolactin concentrations, alter the macronutrient composition of the human milk has been examined in just two studies, one involving domperidone [30], and the other recombinant human prolactin [78]. These studies evaluated changes in a number of breast milk constituents including sodium, calcium, lactose, citrate, protein, fat, alpha-lactalbumin, IgA, and oligosaccharides. Such changes in breast milk composition are particularly important for preterm infants, who have higher protein, energy, mineral, and electrolyte requirements than term infants and are at risk of cumulative nutritional deficits and postnatal growth restriction during early life [79]. As such, multi-component breast milk fortifiers (including protein) are routinely added to expressed breast milk to assist preterm infants in meeting their nutrition and growth requirements. 

Powe et al. investigated the composition of breast milk among 11 women receiving recombinant human prolactin. The mean time following birth was 12 weeks and ranged from 3.5 to 39 weeks). Overall compositional changes mirrored those in women undergoing normal lactogenesis, with an observed increase in lactose and calcium concentrations, and decreased sodium concentrations that all fell within normal ranges [78]. The observed 200 mL/day increase in breast milk production was thought to be related to the increasing synthesis of alpha-lactalbumin and a corresponding increase in lactose levels, which acts as an osmotic agent to draw additional fluid into milk secretory vesicles [78]. Further, greater maturity was reflected by a decrease in milk sodium concentration, consistent with the closure of tight junctions between the mammary epithelial cells. Lastly, recombinant human prolactin administration led to an increase in overall oligosaccharide concentrations [78]. Oligosaccharide concentrations traditionally decrease during lactogenesis, but observed increases in both neutral and acidic oligosaccharide levels could indicate immunological benefits for the infants. In contrast, no statistically significant differences were observed with respect to changes in milk fat, protein or citrate levels.

In comparison to Powe et al., Campbell-Yeo et al. investigated changes in breast milk composition among a cohort of 45 women randomised to either domperidone or placebo. The mean time following birth was not reported, but 75% of women were randomised prior to four weeks. Compared to those receiving placebo, women administered domperidone experienced a greater improvement in breast milk volume of 163 mL/day, with significant increases in breast milk carbohydrate and calcium, but not sodium. While not statistically significant, mean breast milk protein declined by 9.6% in the domperidone group and increased by 3.6% in the placebo group (*p* = 0.16) [30]. The mechanisms contributing to a possible decrease in the protein concentration are unclear, but are comparable to the 15% reduction in protein observed by Powe et al. following the administration of recombinant human prolactin. It is well known that the milk produced by mothers of preterm infants is higher in protein and this becomes comparable to the breast milk produced by mothers of term infants by 10–12 weeks postpartum [80]. It is possible that the administration of galactagogues altering prolactin levels results in a more rapid transition towards mature breast milk with a resultant reduction in the protein concentration occurring as a natural result of increasing breast milk volume. However, given the limited evidence available, future studies evaluating whether galactagogues alter human breast milk composition are required before a more definitive conclusion can be reached. In order to evaluate this effectively, a comparison of breast milk composition must be made to mothers unaffected by lactation insufficiency in order to determine whether any changes in composition are just reflective of improvements in breast milk supply, or the direct effect of galactagogues. Therefore, a key evidence gap relates to the need for a more comprehensive evaluation of breast milk composition that takes into account postnatal age and breast milk volume. 

### 6.3. Predicting Treatment Response to Galactagogues

There are many factors associated with breastfeeding success, with non-pharmacological treatment strategies clearly first-line in the management of lactation insufficiency. Despite adherence to such strategies (e.g., expressing technique, expressing frequency), the treatment response to galactagogues is noted to be variable in clinical practice. Despite this anecdotal experience, we are unaware of any prospective studies that have examined predictors of treatment response to galactagogues such as domperidone. Most clinical studies simply report the average treatment effect of the galactagogue used, but a small number of studies have reported the individual level data with significant variability in treatment response evident. Wan et al. examined changes in daily breast milk volume in response to treatment with either 30 mg or 60 mg daily doses of domperidone. Among the six women studied, two demonstrated no improvement at all in the daily breast milk volume, representing a non-response rate of 33% [27]. A similar treatment response was identified by Da Silva et al. with two of seven women experiencing less than 10% improvement in daily breast milk volume following the treatment with domperidone, representing a non-response rate of 30% [31]. 

Given these findings, understanding factors that contribute towards the response to domperidone is important in identifying groups of women most likely to benefit from treatment, while also enabling the potential identification of other important targets for improving breast milk volume. One such factor may be maternal BMI, where it has already been previously reported that overweight/obese mothers of term infants experience a lower prolactin response to infant suckling [81]. The possible impacts of maternal obesity on treatment response to domperidone have been investigated in a recent retrospective cohort study of mothers of preterm infants. In this study, a significant treatment interaction was observed between domperidone and obesity, with obese women less likely to still be breastfeeding their infants at discharge from the Neonatal Unit than non-obese women, despite both receiving domperidone [82]. Obese women were also noted to be more likely to require a prolonged course of domperidone [82]. This may be due to hormonal or inflammatory differences associated with obesity and warrants further investigation. Of particular interest is the role of the toll-like receptor 4 (TLR4) immune signalling pathway, which plays an important role in obesity and the metabolic syndrome [83]. Further, the TLR4 pathway, regardless of whether it is activated by an infection or a sterile inflammatory pathway, is recognised as a key mediator of preterm labor [84]. This is important as TLR4 signalling may be a key factor involved in the delayed secretory activation and development of low milk supply, commonly experienced by many mothers of preterm infants. When exposed to a bacterial by-product in a mouse model of mastitis [85], it is the TLR4 signalling that is responsible for low milk supply, not the bacterium itself [86]. This surprising finding has led to a new paradigm for lactation insufficiency, whereby TLR4 signalling leads to lactocyte cell death and consequently a reduced capacity to produce milk [87]. The presence of asymptomatic breast inflammation could result in unresponsiveness to domperidone (or other galactagogues) whereby the prolactin-mediated signal to stimulate lactocytes to produce more milk is negated by the inflammatory signal. No studies have prospectively evaluated the role of such inflammatory factors in the development of low milk supply or the response to treatment with galactagogues such as domperidone. 

In addition to previously described factors, serum prolactin levels could determine the treatment response to galactagogues that alter prolactin concentrations. Kauppila et al. demonstrated that three of five mothers who did not respond to the metoclopramide treatment had the highest basal prolactin levels [72]. Further, while the effects of parity on changes in breast milk supply following treatment with domperidone or metoclopramide have not been previously investigated, there is evidence that nulliparous women are more sensitive to the prolactin stimulatory effects of domperidone/metoclopramide than parous women [88]. This could necessitate the need for different doses or treatment approaches based on parity, but requires further investigation.

Other factors that could influence the treatment response to galactagogues include gestational age at birth and timing of the treatment initiation following birth. Both of these were explored in the recent EMPOWER study, which evaluated the use of domperidone compared to placebo over 14 days [37]. Following the end of the initial treatment period women who received placebo went on to receive domperidone for 14 days. This enabled investigation of the potential effects of delaying the treatment with domperidone for 14 days. At the end of 28 days, the mean daily milk volume was similar between those who immediately started on domperidone compared with those who had treatment with domperidone delayed by 14 days (290 ± 211 mL/day vs. 302 ± 230 mL/day; *p* = 0.88) [37]. Additionally, they observed no significant difference between the number of women who had delivered between 23–26 weeks gestation or 27–29 weeks gestation and achieved a 50% increase in breast milk volume (72.9% compared to 64.2%; *p* = 0.38) [89].

## 7. Conclusions

The largest and strongest body of evidence supports the use of domperidone as the first-line medication for the pharmacological management of lactation insufficiency but the observed treatment effects are modest at best. There is currently insufficient evidence to determine whether the use of galactagogues has any negative impacts on the micro- and macro-nutrient composition of breast milk and this requires further research. There is emerging evidence of potential differences in the treatment response to galactagogues and further research in this space is critical for improving the therapeutic management of lactation insufficiency. There is a critical need for studies identifying underlying mechanisms contributing towards the development of lactation insufficiency and how these impact the treatment response to various pharmaceutical galactagogues. Such knowledge would facilitate the development and evaluation of novel pharmaceutical treatments for lactation insufficiency while also improve our understanding of individual variability in treatment response to improve utilisation and outcomes of existing treatments. 

## Figures and Tables

**Figure 1 nutrients-11-00974-f001:**
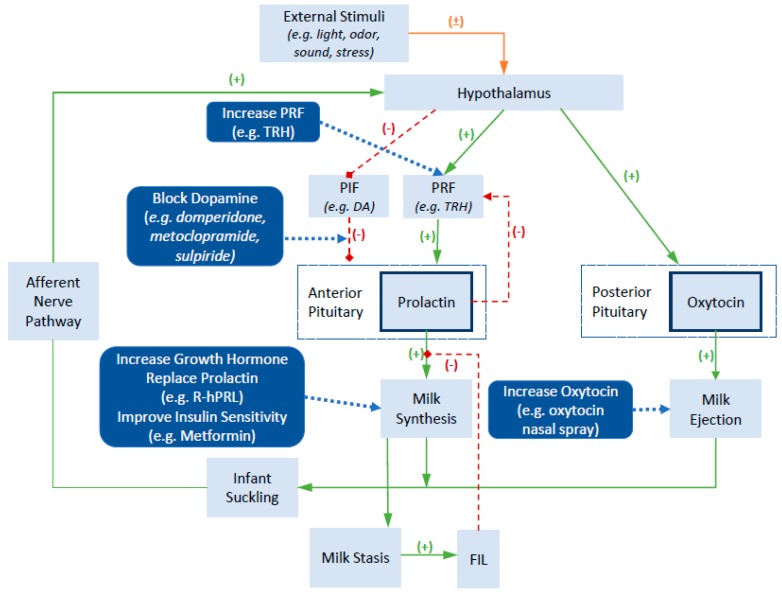
Physiology of lactation and mechanisms of action of various pharmaceutical galactagogues (Adapted from [14,15,16]). Abbreviations: TRH, thyrotrophin releasing hormone; PIF, prolactin inhibitory factor; PRF, prolactin releasing factor; DA, dopamine; R-hPRL, recombinant human prolactin; FIL, feedback inhibitor of lactation.

**Table 1 nutrients-11-00974-t001:** Overview of controlled studies evaluating the efficacy of pharmaceutical galactagogues for the treatment of lactation insufficiency.

Study (Country)	Intervention (n/N) and Dose	Duration (Days)	Delivery Gestation (Weeks)	Infant Age (Weeks)	Effect on Milk Production
**Domperidone**					
Asztalos 2017 (Canada) [37]	*Domperidone* (45): 10 mg TDS *Placebo* (40/45)	14	<30	>1 to ≤3	Change in milk supply: *Domperidone*: 121 (±96) to 267 (±189) mL/day; *Placebo*: 115( ±95) to 217 (±168) mL/day; *p* = 0.20
Campbell-Yeo 2010 (Canada) [30]	*Domperidone* (21/22): 10 mg TDS *Placebo* (24)	14	<31	≥3	Change in milk supply: *Domperidone*: 184 (±167) to 380 (±202) mL/day; *Placebo*: 218 (±155) to 251 (±172) mL/day
da Silva 2001 (Canada) [31]	*Domperidone* (6/11): 10 mg TDS *Placebo* (8/9)	7	<37 †	Any	Difference from baseline: *Domperidone*: 49.5 (±29.4) mL/day; *Placebo*: 8.0 (±39.5) mL/day; *p* < 0.05
Fazilla 2017 (Indonesia) [32]	*Domperidone* (25): 10 mg TDS *Placebo* (25):	7	<37	>1	Difference from baseline: *Domperidone*: 181.6 (±80.2) mL/day; *Placebo*: 72.4 (±57.8) mL/day; *p* = 0.0001
Inam 2013 (Pakistan) [33]	*Domperidone* (50): 10 mg TDS *Placebo* (50)	7	Term	1	Number reaching target of ≥ 50 mL per single expression: *Domperidone*: 36/50 (76%); *Placebo*: 11/50 (22%); *p* = 0.002
Knoppert 2013 (Canada) [34]	*Domperidone* (5/8): 10 mg TDS *Domperidone* (6/7): 20 mg TDS	28	<33	≥2 to ≤3	Change in milk supply: 10 mg TDS: 150 to 420 mL/day; 20 mg TDS: 300 to 720 mL/day
Petraglia 1985 (Italy) [35]	*Domperidone* (9): 10 mg TDS *Placebo* (8)	10	Term	>2	Change in milk supply: *Domperidone*: 347 (±36) to 673 (±44) mL/day; *Placebo*: 335 (±30) to 398 (±45) mL/day; *p* < 0.01
Rai 2016 (India) [36]	*Domperidone* (14/16): 10 mg TDS *Placebo* (16)	7	<37†	≥1 to ≤2	Difference from baseline: Domperidone: Median 186.0 (IQR 126.5 to 240) mL/day; Placebo: Median 70 (IQR 49.5 to 97); *p* = 0.004
Wan 2008 (Australia) [27]	*Domperidone* (4/7): 10 mg TDS & 20 mg TDS	7–14	<37	>2	Change in milk supply: Baseline: 208.8 (±182.3) mL/day; 30 mg: 566.4 (±229.3) mL/day; 60 mg: 705.6 (±388.1) mL/day
**Metoclopramide**					
Kauppila 1981 (Finland) [39]	*Metoclopramide* (37/45): 5 mg TDS, 10 mg TDS, 15 mg TDS & *placebo*	14	Term	>1	Difference from baseline (single feed): 5 mg TDS: 11.2 ± 28.1 mL; 10 mg TDS: 42.5 ± 34.7 mL; 15 mg TDS: 50.0 ± 35.9 mL; *Placebo*: 4.0 ± 27.5 mL
Kauppila 1985 (Finland) [38]	*Metoclopramide* (8): 10 mg TDS *Placebo* (5)	21	Term	≥4 to ≤20	Change in milk supply: *Metoclopramide*: 285 (±75) to 530 (±162) mL/day (*p* < 0.01); *Placebo*: individual results not stated
Sakha 2008 (Iran) [40]	*Metoclopramide* (10): 10 mg TDS *Placebo* (10)	15	Term	NR	Infant weight gain: *Metoclopramide*: 328.5 g; *Placebo*: 351.5 g (*p* = 0.68)
**Domperidone/Metoclopramide**					
Blank 2000 (Australia) [28]	*Domperidone* (9): 10 mg TDS *Metoclopramide* (11): 10 mg TDS *Placebo* (9)	5	<34	≥1	Change in milk supply: *Domperidone*: 120 (±81) to 239 (±105) mL/day; *Metoclopramide*: 100 (±53) to 184 (±100) mL/day; *Placebo*: 143 (±57) to 172 (±117) mL/day
Ingram 2012 (Canada) [29]	*Domperidone* (31/38): 10 mg TDS *Metoclopramide* (34/42): 10 mg TDS	10	<37†	NR	Change in milk supply: *Domperidone*: 174 (±126) to 285 (±158) mL/day; *Metoclopramide*: 133 (±115) to 212 (±154) mL/day; *Difference*: 31.0 (−5.67 to 67.6) mL/day
**Sulpiride**					
Ylikorkala 1982 (Finland) [41]	*Sulpiride* (14): 50 mg TDS *Placebo* (12/14)	28	NS	≤16	Difference from baseline: *Sulpiride*: 265 mL/day; *Placebo*: −50 mL/day
Ylikorkala 1984 (Finland) [42]	*Sulpiride* (13/14): 50 mg TDS *Placebo* (11/14)	14	NS	≤16	Difference from baseline: *Sulpiride*: 646 (±67) mL/day; *Placebo*: 428 (±71) mL/day (*p* < 0.05)
**Human Growth Hormone (hGH)**					
Gunn 1996 (New Zealand) [43]	*hGH* (9/10): 0.2 IU/kg/day SC (max 16 IU/kg/day) *Placebo* (9/10)	7	26–34	NR	Change from baseline: *hGH*: 139 (±49) to 175 (±46) mL/day; *p* < 0.01 *Placebo*: 93 (±50) to 102 (±69) mL/day; *p* = NS
Milsom 1998 (New Zealand) [44]	*hGH* (5): 0.05 IU/kg/day SC *hGH* (5): 0.1 IU/kg/day SC *hGH* (6): 0.2 IU/kg/day SC	7	Term	<16	Percentage increase in breast milk volume: High dose group (0.2 IU): 36 ± 12.6 Low/Mid dose group (0.05/0.1 IU): 4.7 ± 9.7; *p* < 0.04
**Recombinant Human Prolactin (R-hPRL)**					
Powe 2010 (USA) [45]	*R-hPRL* (3): 60 mcg/kg SC BD *R-hPRL* (3): 60 mcg/kg SC daily *Placebo* (4)	7	24–32	≥1 to ≤4	Percentage change in milk supply: *R-hPRL* (Twice Daily): 429 (±338%); *R-hPRL* (Once Daily): 44 (±28%); *Placebo*: −12 (±27%)
**Thyrotrophin-Releasing Hormone (TRH)**					
Zarate 1975 (Mexico) [46]	*TRH* (5): 20 mg TDS PO *Placebo* (4)	7	NR	NR	No change in milk production (actual volume not reported)
Peters 1991 (Germany) [47]	*TRH* (10): 1 mg QID IN *Placebo* (9)	10	Term	Day 6	Change from baseline: *TRH*: 142.0 (±33.9) to 253.0 (±105.3) g/day (*p* = 0.014); *Placebo*: 150.0 (±46.2) to 140.6 (±57.7) g/day (*p* = 0.87)
**Metformin**					
Nommsen-Rivers 2019 (USA) [48]	*Metformin* (10): Day 1–7 – 750 mg/day; Day 8–14 – 1500 mg/day; Day 14–28 – 2000 mg/day *Placebo* (5)	28	Term	1–8	Difference from baseline: *Metformin*: Median 8 (IQR −23 to 33) mL/day; *Placebo*: Median −58 (IQR −62 to −1) mL/day

Abbreviations: TDS, three times daily; BD, twice daily; SC, subcutaneous; IN, intranasal; QID, four times daily; NR, not reported. † denotes infants admitted to neonatal unit.

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
