# Peer review of "What Evidence Do We Have for Pharmaceutical Galactagogues in the Treatment of Lactation Insufficiency?—A Narrative Review"

_nutrients, 2019, doi:10.3390/nu11050974_

Reviewer 1 Report

This is a non-systematic review paper on Pharmacological galatagoges, exploring doperidone, metaclopromaide, prolactin oxytocin, metformin.

The writing is very good and the content is comprehensive and persuasive. However, there are some minor editing errors in the References section, possibly due to the use of Reference manager (australian, british, empower trial, not capitialized for example).

 Suggestions: 

1. Convert this paper to a traditional structure: Introduction, Methods, Results, and Discussion outline. Omit paragraph numbers. Headings are fine. Consider Including the effects of mode of delivery and early skin to skin contact on lactation in the Introduction. 

2. State at the last paragraph of the Introduction that you have published a meta-analysis of domperidone. Then state the aims of this paper.

3. Insert a Methods section. How did you do your search? Outline the exclusion criteria, How did you assess the quality of the data etc.

4. Table 1 could be in the Results section.

Author Response

Comment 1: This is a non-systematic review paper on Pharmacological galatagoges, exploring doperidone, metaclopromaide, prolactin oxytocin, metformin.

Response 1: Correct, this is a non-systematic review – we have added ‘narrative review’ to the title to avoid any potential confusion.

Comment 2: The writing is very good and the content is comprehensive and persuasive. However, there are some minor editing errors in the References section, possibly due to the use of Reference manager (australian, british, empower trial, not capitialized for example).

Response 2: We have amended the references to ensure correct use of capitals.

Suggestions: 

Comment 3: Convert this paper to a traditional structure: Introduction, Methods, Results, and Discussion outline. Omit paragraph numbers. Headings are fine. Consider Including the effects of mode of delivery and early skin to skin contact on lactation in the Introduction. 

Response 3:  As previously stated this is a narrative review and we have chosen a slightly different format than that used for traditional systematic reviews. The current format is consistent with previous narrative reviews published in the journal ‘Nutrients’ and we utilised these previous reviews in formatting this manuscript (e.g. https://www.mdpi.com/2072-6643/10/5/600).

Comment 4: State at the last paragraph of the Introduction that you have published a meta-analysis of domperidone. Then state the aims of this paper.

Response 4: As previously stated, we do not feel that changing the structure of this narrative review is necessary. We also do not feel that referencing our previously meta-analysis on domperidone in the introduction is necessary nor valuable for the reader. This review is focused on the use of any galactagogues to increase breast milk supply, rather than domperidone alone. Our previous domperidone meta-analysis only included 5 studies and was restricted to studies undertaken in mothers of preterm infants, whereas this review includes 22 studies and is much more exhaustive. We have cited our previous meta-analysis in the discussion related to the efficacy of domperidone and feel that this is sufficient.

Comment 5: Insert a Methods section. How did you do your search? Outline the exclusion criteria, How did you assess the quality of the data etc.

Response 5: As previously noted, this was a narrative review, rather than systematic review. That said, we have included some additional details regarding how literature was searched and obtained for this review:

“In order to evaluate the efficacy and safety of galactagogues wesearched four electronic databases from inception to January 2019: Ovid Medline, EMBASE, Web of Science, and SCOPUS.Medical subject headings (e.g. MeSH headings) and free word combinations using Boolean logic of the following search items were used: ‘galactagogues’ORindividual medication names (e.g.‘domperidone’) AND ‘lactation’ OR ‘low milk supply’. Previous reviews, bibliographies of published trials and cross references were also searched. No language restrictions were applied.Studies were restricted to those investigating the use of galactagogues in the treatment of established lactation insufficiency.”

Comment 6: Table 1 could be in the Results section.

Response 6: Given there are no changes to the structure of the manuscript Table 1 remains where is previously was.

Reviewer 2 Report

Thank you for the opportunity to review with extremely well-written review of pharmaceutical galactagogues.  While not imperative, the review would be stronger if it followed PRISMA guidelines. In the first section, it is necessary to clarify you are referring to mother’s own milk when you state breast milk (this could refer to either to either donor human milk or mother’s own milk).

Page 2: line 239: additional information regarding the risk of VA in non-lactating adults would be helpful.  For example what conditions are treated with this medication and at what dose is generally provided.

Page 3: line 305: while the possible risk of crossing the blood brain barrier of the mother is discussed, the potential risk to the infant is not included.

Page 4 line 338: are there any potential side effects to Sulpiride?

Page 7 section 6.2. A physiologic mechanism regarding why galactogogues would change milk composition is needed.

Author Response

Comment 1: Thank you for the opportunity to review with extremely well-written review of pharmaceutical galactagogues.  While not imperative, the review would be stronger if it followed PRISMA guidelines. In the first section, it is necessary to clarify you are referring to mother’s own milk when you state breast milk (this could refer to either to either donor human milk or mother’s own milk).

Response 1: The PRISMA guidelines cover systematic reviews and meta-analyses. As previously described, this is a narrative literature review and so the guidelines are not relevant. We have added ‘narrative review’ to the title and abstract to make this point clear.

Comment 2: Page 2: line 239: additional information regarding the risk of VA in non-lactating adults would be helpful.  For example what conditions are treated with this medication and at what dose is generally provided.

Response 2: Further detail has been provided in this section:

“Domperidone was initially developed as an antiemetic and prokinetic, with previous recommended doses ranging from 30 mg/day to60mg/day or greater[65].Some observational studies suggested that the risk of VA or sudden cardiac death may be increased if daily doses were greater than 30 mg or if patients weremale orgreater than 60 years old[60,61].More detail regarding domperidone and QT-interval effects among non-lactating adults can be found elsewhere[60,61].”

Comment 3: Page 3: line 305: while the possible risk of crossing the blood brain barrier of the mother is discussed, the potential risk to the infant is not included.

Response 3: Discussion of potential risks to the infant is provided in the previous paragraph (see below). We have added a sentence to outline potential infant concerns relating to potential for metoclopramide to cross the blood brain barrier:

“Notably, metoclopramide was detected in the plasma of one infant and while no short-term neonatal adverse events have been identified, whether infants are more susceptible to potential longer-term adverse events of metoclopramide is unknown.This is of concern given the ease with which metoclopramide crosses the blood-brain barrier and therefore the potential to interrupt dopamine signaling in the newborn.”

Comment 4: Page 4 line 338: are there any potential side effects to Sulpiride?

Response 4: We have added a paragraph to this section outlining potential maternal and infant side effects associated with the use of sulpiride:

“While previous clinical trials in lactation have reported few adverse events associated with the use of sulpirideand the doses used in these trials are lower than thosethought to produce significantneuroleptic effects in adults,data are still limited and there remainconcerns regarding untoward maternal or infant effects. Common adverse events include sedation and weight gain, while sulpiride may also case extrapyramidal side effects similar to metoclopramide.Of greater concern is that the relative infant exposure to sulpiride through breast milk is up to 20% of maternal weight-adjusted dose[47], which is much greater than for other galactagogues.”

Comment 5: Page 7 section 6.2. A physiologic mechanism regarding why galactogogues would change milk composition is needed.

Response 5: We have added a paragraph to this section to provide biological plausibility:

Prolactin is thought to mediate changes in breast milk composition occurring during normal lactogenesis[78]. In vitro and animal studies provide evidence that in early lactation prolactin promotes closure of epithelial tight junctions between alveolar cells and increases synthesis of α-lactalbumin, which in turn increases breast milk volume[78]. Prolactin has also been demonstrated to influence secretion of immunomodulatory factors in breast milk, with the decreasing concentrations of these factors occurring over the course of lactation mirroring reductions in serum prolactin concentrations also occurring over time[78].Whether galactagogues, particularly those altering prolactin concentrations,alter the macronutrient composition of human milk has been examined in just two studies, one involving domperidone[30], and the other recombinant human prolactin[78].”